# A Historical Misconception in Clinical Trials of Drugs for Cancer—Age Grouping

**DOI:** 10.3390/jpm12121998

**Published:** 2022-12-02

**Authors:** Jingyu Chen, Lan Yao, Abdulmohsin J. Alamoudi, Lotfi Aleya, Weikuan Gu

**Affiliations:** 1Department of Chinese Medicine Internal Medicine, Xiyuan Hospital, China Academy of Chinese Medical Sciences, No. 1, Xiyuan Playground, Haidian District, Beijing 100091, China; 2Department of Nutrition and Health Science, College of Health, Ball State University, Muncie, IN 47306, USA; 3Department of Orthopedic Surgery and BME, College of Medicine, University of Tennessee Health Science Center, Memphis, TN 38163, USA; 4Department of Pharmacology and Toxicology, Faculty of Pharmacy, King Abdulaziz University, Jeddah 21589, Saudi Arabia; 5Chrono-Environnement Laboratory, UMR CNRS 6249, Bourgogne Franche-Comté Université, CEDEX 21010, F-25030 Besançon, France; 6Research Service, Memphis VA Medical Center, 1030 Jefferson Avenue, Memphis, TN 38104, USA; 7Department of Pharmaceutical Sciences, College of Pharmacy, University of Tennessee Health Science Center, 881 Madison Ave, Memphis, TN 38163, USA

**Keywords:** cancer, clinical trial, life stage, age, patients

## Abstract

In clinical trials of cancer drugs, grouping by age is a very common grouping method, as it can allow for a visual comparison of the different pharmaceutical responses in patients at different age stages. Under the guidance of this thinking, many researchers use age grouping when studying clinical cancer drugs. However, even people at the same age may be at different stages in their lives, such as individuals who are going through puberty, menopause/andropause, or intermediate transition, as well as childhood and old age, affected by factors such as hormone levels, immune responses, ethnic groups, and regions. Every individual has different cancer symptoms and responses to drugs; therefore, the experimental effect of life stage grouping will be more obvious and clearer. Not only does this conclusion apply to cancer drugs, but it also applies to clinical trials for other diseases. In addition, this does not mean that age grouping should be completely abandoned. Life stage is a more general interval that can be further divided into life stage groups according to the age of the patients. Based on the principal law of lifespan (PLOSP), age trends in life stages also need to be updated from time to time. To date, life stage grouping has not been discussed systematically and has not been used as a grouping method for cancer patients. In this paper, life stage grouping is discussed as one of the important grouping categories in cancer clinical trials.

## 1. Introduction

Since the first cancer clinical trial, grouping patients has been essential. Subjects are grouped because there are great diversities in drug treatment response. These diversities exist not only in disease stages, genomic variations, and sex but also in different life stages. Thus, the effect of the same drug may vary among populations of different groups. Understanding the efficacy of a drug in various population groups provides critical guidance on its therapeutic application among such groups.

Dividing patients into different categorical groups is a complicated task. It is easy to divide some categories, such as sex and ethnic groups, while it is difficult and time-consuming to divide others, such as disease phenotypes, including disease stages and pathological characterizations. Other patient characteristics, such as smoking status and drinking habits, are grouped based on a consensus among investigators and the public. Grouping by age appears to be relatively easy, but confusion exists, because the ages of patients in the same drug category tend to be grouped differently in various clinical trials. As personalized medicine depends heavily on the categorization of individuals into the right groups, determining how to group patients with cancer by age in a scientific manner has become an important issue that needs to be solved.

Despite the differences in age grouping, the majority of clinical trials do not group patients based on life stage [1,2]. This article discusses the need to categorize patients with cancer based on their life stage.

## 2. Life Stage, Age, and Cancers

### 2.1. Age-Specific Cancers

Overall, cancer incidence rates increase with age before 85, with a few cancers occurring at a young age [3]. Thus, in the long term, the incidence of some cancer types occurs in a bell shape. It is known that some cancers are more common in children, such as acute lymphocytic leukemia (ALL), bone sarcomas, and rhabdomyosarcoma. Unfortunately, there has been no systematic investigation into the relationship between life stages and the incidence of cancer types.

Lymphocytic leukemia occurs in almost all age groups. However, there is a peak in its incidence rate during childhood, followed by a low incidence rate before another high incidence rate after the age of 50 years. The incidence rate runs in the opposite direction to the hormone level (Figure 1). Thus, it is most likely that the incidence rate is low during the period between puberty and menopause, when the immune response is high. However, the incidence of lymphocytic leukemia has been recorded in various age groups. Whether the incidence of lymphocytic leukemia is more likely related to life stage or age is a critical question that needs to be answered in the future.

However, age at the time of breast cancer diagnosis is around 50 years [4,5]. Bae examined the incidence rates of breast cancer by age in Korean women over ten years between 1993 and 2002 [4]. The data showed that the peak appears in the age range of 45–49 years and declines thereafter. Because the menopausal transition most often begins between the ages of 45 and 55, one may wonder whether the menopausal transition impacts the occurrence of breast cancer. A report based on this indicated that the time intervals between reproductive events had different effects on breast cancer outcomes depending on menopausal status [6].

For prostate cancer, there are multiple reports on the incidence rates of various age groups. Among men, the age groups of 45–64, 50–69, and 45–69 years have been reported to have increased incidence rates, while the age group of 75–84 years has also been reported to have an increased incidence rate [7]. The link between the incidence of prostate cancer and andropause has not been reported to date, although studies clarifying such a connection are necessary.

Understanding whether these cancers are linked to specific life stages will benefit their prevention and treatment. If a particular type of cancer is linked to childhood before puberty, the mechanism related to low hormone levels can then be investigated, and prevention measures can be implemented. Drugs related to life stage pathways can be developed for the proper treatment at different life stages.

### 2.2. Age Grouping in Clinical Trials

The age of a patient is easily obtained and essential information in all clinical trials. However, the grouping of patients by age varies across clinical trials. The fact is that, in most cases, the average age of the patient population in the trial determines the age grouping. The critical issue is that, in the majority of clinical trials, no solid scientific bases for the age-based grouping are given. Table 1, Table 2 and Table 3 provide examples of phase 3 clinical trials to show how the patients were grouped. These examples show grouping based on age in patients with lung cancer, breast cancer, and lymphoblastic leukemia.

In the trials of drugs for the treatment of lung cancer, age was divided into a variety of groups (Table 1) [8,9,10,11,12,13,14,15,16,17,18]. While the majority of the studies grouped patients into two groups, one group comprising those aged ≥65 and the other comprising those aged <65, others grouped patients by different ages and employed even more age groups. For example, in a study of drug effects on squamous non-small-cell lung cancer, Kogure et al. divided patients into two age groups of ≥75 and <75 [10]. In a study of small-sized peripheral non-small-cell lung cancer, Saji et al. also divided patients into the same two groups [15]. In a study conducted by Hellmann et al., patients were divided into three age groups of ≥75, 65–75, and <65 [18].

In the clinical trials for the treatment of breast cancer, age grouping showed great diversity (Table 2) [19,20,21,22,23,24,25,26]. In a trial with patients with non-low-risk ductal carcinoma in situ in the breast, where the median age of the patients was 58, the authors [19] divided patients into four age groups: ≥70, 60–69, 50–59, and <50. In a trial with early-stage breast cancer, Del Mastro et al. [23] also divided patients into four age groups but with different age ranges: ≥76, 65–75, 55–64, and <55. Several studies divided patients into two groups (Table 2) [20,21,24,25]. However, the age groups were different when the ages of the patients were similar. For example, in a clinical trial with patients with metastatic triple-negative breast cancer, the median age was 50 [20], and the authors divided the patients into two groups: one comprising those aged ≥40 and the other comprising those aged < 40. In another clinical trial with patients with ERBB2-positive breast cancer, the patients had a median age of 50 [24]. However, the patients were divided into two groups: those aged ≥50 and those aged <50. In a study in patients with early breast cancer, with the median age of 52, the patients were divided into two groups: those aged ≤50 and those aged >50 [25].

Similar to those of breast cancer, the age groups in the clinical trials of lymphoblastic leukemia were mixed, and they are difficult to explain (Table 3) [27,28,29,30,31,32,33]. In the most recent study of acute lymphoblastic leukemia, Yang et al. [27] divided patients into three age groups: ≥10, 1–10, and <1. Two early studies divided patients into two groups: one with age groups of ≥10 and 1–9 [32], and the other with age groups of ≥10 and <10 [33]. Furthermore, Peters et al. [31] divided patients into four age groups of >14, >10–14, >6–10, and 4–6. In a study of high-risk B-lymphoblastic leukemia, Burke et al. [28] divided patients into three age groups of 16–30, 10–15, and 1–9. In a study of the first relapse of B-cell acute lymphoblastic leukemia, Brown et al. [30] divided patients into five age groups of 21–27, 18–20, 13–17, 10–12, and 1–9.

### 2.3. The Disagreement between Age and Life Stage

While we do not have questions regarding grouping by age, we want to examine the bases of age grouping. The most important issue in the grouping of patients by different ages is that age is not equal to life stage [2]. Although every human being goes through puberty and menopause/andropause, the age at which one goes through these life stages varies greatly among individuals, ethnic groups, regions, and countries. A female at the age of 60 years may have gone through menopause, while another female at the same age may not have gone through menopause. The same can be said for two men: one may have gone through andropause, while the other may not have gone through andropause. Regarding young patients, a 12-year-old patient may or may not have gone through puberty. Thus, the previous instances of age-based grouping does not represent the differences between life stages.

One of the recent theories on the extension of the human lifespan is the principal law of lifespan (PLOSP). According to the PLOSP, the same strategy can have dramatically different or even opposite effects on different life stages. In the case of drug treatment for girls aged 12 years, the effect of a drug before puberty will be different from that after puberty.

### 2.4. The Difference in Response to Cancer Drug Treatment between Different Life Stages Is More than the Difference between Arbitrarily Divided Age Groups

It is well-known that physiological and immunological conditions differ tremendously before puberty and after puberty [2]. The immune system of a person who has gone through menopause is weaker than that of a person who has not gone through menopause [2]. However, the difference between age-based groups may not be as large as that between life-stage-based groups. For example, if a clinical trial groups female patients into two groups, one comprising those aged ≤50 and the other comprising those aged >50, the difference in responses to an immune system drug will not be the same as that between female patients who have not undergone menopause and those who have undergone menopause. Because the menopausal transition most often begins between the ages of 45 and 55 years, within the group of those aged ≤50, there will be individuals who have not undergone menopause, those undergoing menopause, and those who have undergone menopause. Similarly, in the group of those aged >50, there also will be individuals meeting the criteria of these three groups. In such a case, the degree of the immune response before, during, and after menopause in these two groups will be similar. Instead, if the patients are divided into three groups based on the stages of menopause, it is most likely that there will be a considerable difference in the immune response among these three groups.

In addition, in many cases, the age ranges in clinical trials fluctuate considerably. For example, in a recently reported clinical trial in breast cancer [34], while the median ages of the treatment and control populations were 52 and 51, respectively, the age ranges were from 23 to 74. Within such a population of mixed ages, the responses to the drug treatment at different life stages may vary greatly. Obviously, grouping by life stage may be helpful to identify the differences in the responses to drug treatment among various groups.

### 2.5. Lack of Study on Drugs of Age Specificity

There is a lack of studies on the life stage specificity of drugs not only for cancers but also for other diseases. When reviewing the impact of age on antiretroviral drug pharmacokinetics in the treatment of adults living with HIV, Calcagno et al. [35] found that the available evidence of a potential detrimental effect in elderly people living with HIV is limited by study design and small sample sizes. Therefore, no definitive suggestions for the utilization of antiretroviral drugs in elderly patients have been made.

Regarding drug pathways, we did not find a study that systematically investigated the differences in drug effects on the different life stages. However, in a study on the expressions of drug transporters in human kidneys, Joseph et al. [36] reported that the expression levels of SLC22A2, SLC22A12, SLC6A16, and ABCB6 were significantly higher in females aged <50 years than in females aged ≥50 years. These rare reports of age specificity indicate that more investigations into the differences in life stages are essential for the personalized utilization of drugs.

### 2.6. Current Considerations for Cancer Treatment of Different Ages

For children, the current considerations for cancer treatment are methodologies that are better for them [37,38]. We do not see a clear definition for children in children cancer centers or on the labels of cancer drugs. For the dosage of many drugs, the labels usually indicate certain ages, such as under 12 years.

For older adults, the consideration is that older adults are more likely to have chronic health conditions, such as diabetes and heart disease, or that older adults are more likely to have serious side effects from chemotherapy [39]. On most clinical websites, numerous factors are listed for the consideration of older patients, while life stages, such as menopause and andropause, are not listed.

### 2.7. Hormone Levels and Life Stages

It is well-known that the levels of hormones are associated with different life stages in both sexes [2]. In general, hormone levels increase during the life stage of body growth, maintain a relatively high level during the reproductive stage, and decrease during the aging stage. It is also known that the level of hormones is associated with breast cancer and the response to the drug treatment of some cancers [22,40,41]. As we discussed above, in many cases, age grouping does not agree with life stages. Therefore, based on the relationship between the levels of hormones, disease development, and the response of patients with cancer to drug treatment, grouping by life sage will benefit cancer treatment.

### 2.8. Immune System and Life Stages

Similar to hormone levels, the immune system develops with body growth, keeps functioning well during the reproductive stage, and significantly decreases its activity when entering the aging stage. Immune levels are directly related to cancer development and treatment [1,2]. The immune response and physiological status are significantly different across the life stages. Thus, the immune system is highly relevant to the determination of treatment options and drug dosages of patients with cancer. Life stage grouping can correctly reflect how a patient with cancer responds to treatment.

## 3. Cancer Therapy Design and Life Stages

Cancer treatment is slowly advancing from the classical use of nonspecific cytotoxic drugs targeting the generic mechanisms of cell growth and proliferation to individualized and cancer-specific treatment. However, to date, every approach has been practically used in clinics. By examining each drug category of cancer chemotherapy, the importance of life stages emerges.

### 3.1. Body Growth and the Early Drugs of Chemotherapy

Cancer drugs were discovered due to the alkylating activity of nitrogen mustard [42,43]. All alkylating agents, including nitrosourea compounds, alkyl sulfates, ethyleneimine derivatives, epoxides, triazene compounds, and metal salts, kill cancer cells because they prevent the cell from reproducing via cross-linking strands of DNA, particularly at the N-7 position of guanine [42]. However, they have a common problem: they kill healthy cells while destroying cancer cells because they are nonspecific for cell type and the cell cycle phase. In this case, a person at the body growth stage is different from a person in the reproductive stage and a person in the aging stage. Puberty is a better marker than age when considering the cell-dividing activity of an individual. Such a difference exists not only for alkylating agents but also for agents with similar mechanisms of action, such as antimetabolites, antimitotics, polyamine inhibitors, and iron-modulating drugs.

### 3.2. Life Stages and Targeted Therapy

Targeted therapy for cancer treatment targets proteins that control how cancer cells grow, divide, and spread. Currently, the majority of the drugs used are tyrosine and serine/threonine protein kinase inhibitors and monoclonal antibodies. While these selective inhibitors are able to act mainly on cancer cells, they still have minor side effects on healthy cells. The effects vary in the different life stages or growth stage. Furthermore, the expression and activity of tyrosine and serine/threonine protein kinase are influenced by growth hormones [44]. Therefore, life stage is an important factor that should be considered in the application of targeted therapy.

### 3.3. Life Stages and Immune Checkpoint Inhibitors

Cancer immunotherapy, such as anti-programmed cell death protein 1 antibody (anti-PD1), and monoclonal antibodies, such as anti-cytotoxic T-lymphocyte-associated antigen 4 (anti-CTLA4), directly work on tumor antigens [45]. The inhibition of tumor antigens enhances the immune response so that cancer cells can be killed or eliminated. The key issue is that the level of the human immune response is directly related to life stage [2]. The dosage for the stimulation of the immune response is influenced by life stage. Carefully determining and utilizing drugs based on an individual’s life stage will increase the success rate of immunotherapy.

### 3.4. Life Stages and Molecular Radiotherapy

Age is known to be a factor that influences molecular radiotherapy [46]. Because the growth rates and hormone levels at different life stages are different, the side effects on and the damage to healthy tissues are different. Plus, the immune response determines the capability and speed of repairing the damage caused by radiation to healthy cells and tissues. Considering life stage as a factor, along with age, is necessary.

## 4. Grouping According to Life Stage and Detailed Analyses

While we feel that using life stage as a grouping method in cancer clinical trials is critical to correctly evaluate the response of patients to treatment, we are not ignoring age grouping. Age groups can further divide within each life stage. In particular, if the study population has patients mostly aged over 65 years, post-menopause and/or post-andropause groups can be further divided into different groups based on age. Thus, detailed age-based grouping can provide subgroups for each life stage. The other reason why incorporating life stage grouping is better than only incorporating age grouping is that the ages at the times of life stage transition in the human population are changing. In the past half century, the average age of puberty has decreased by around two years, at least in developed countries. In the opposite way, the age of menopause/andropause has increased on average by around 3 years [1]. The efficacy of a drug for the treatment of a cancer in a 12-year-old girl may be different after tens of years, as a girl of the same age may have already undergone puberty.

## 5. Conclusions

Life stages can be used in clinical trials and cancer treatment. Figure 2 describes the possible route to promote the use of life stages in clinical trials and therapeutic applications.

### 5.1. Basic Research to Understand Similarities and Differences in Cancer Development and Differences among Life Stages

Basic research can identify the differences at molecular levels in cancer development and treatment among different life stages. By comparing the whole-genome expression profiles of healthy and cancer tissues among the growth stage before puberty, after puberty, and after menopause, the similarities and differences in a variety of molecular pathways can be determined. Furthermore, gene expression profiles in different organs or tissues may vary at different life stages. Comparisons can also be made with the life stage transitional period, for example, comparing the periods of puberty and menopause to those before and after. Such a detailed analysis will reveal the molecular mechanisms that drive the difference in cancer development among the different life stages.

### 5.2. Preclinical Test of the Similarities and Differences Using Animal Models

Cancer drug tests with animal models have not been divided into life stages. When testing new drugs using animal models, an examination of the response of the animals to the treatment at different life stages is important in order to determine whether a drug is life-stage-specific or to what extent the variation in the effect will be if it is used at different life stages. Not every drug will be life-stage-specific. The results from animal models will provide information for consideration when the drug is tested in the human population.

### 5.3. Standardization of Life Stage Grouping in Drug Test Protocols

The life stages of patients in clinical trials should be written into the guidelines for clinical trials, such as the NCI Criteria Guidance and the FDA clinical trial guidance [47,48]. Changing guidelines is not an easy task, but it should eventually happen. Changes need public support and consensus of the scientific community. We believe that soon, due to the new supporting data from basic research, as well as evidence from clinical data, such a change will occur.

## Figures and Tables

**Figure 1 jpm-12-01998-f001:**
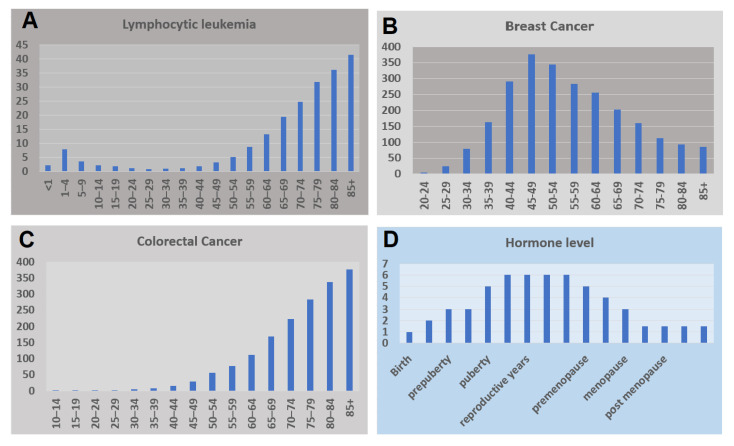
Distribution of incidence rates and hormone levels. (**A**). The disease incident rate of Lymphocytic leukemia at different ages. (**B**). The disease incident rate of Breast cancer at different ages. (**C**). The disease incident rate of Colorectal cancer at different ages. (**D**). The levels of hormone at different life stages.

**Figure 2 jpm-12-01998-f002:**
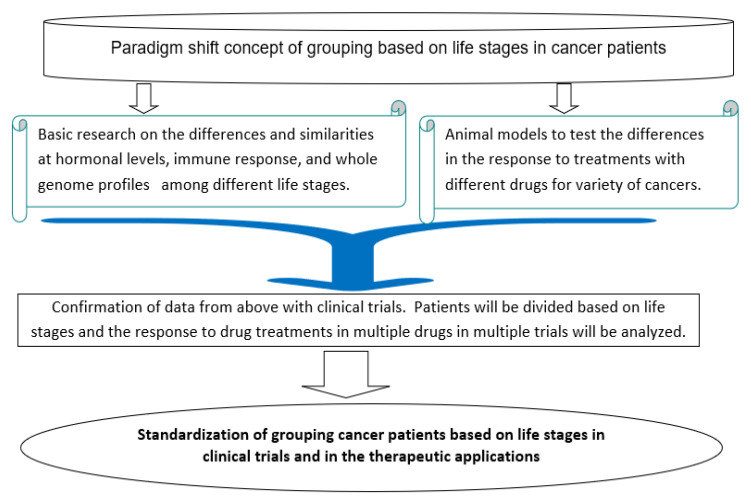
Possible route to promote the use of life stages in clinical trials and therapeutic applications.

**Table 1 jpm-12-01998-t001:** Age grouping in clinical trials of drugs for lung cancer.

First Author/Reference	Disease	Age Groups in Treament Population						Age Groups in Ontrol/Other Treatment Population					
		Median	≥75	<75	65–75	<65	≥65	Median	≥75	<75	65–75	<65	≥65
Cheng [8]	Extensive-Stage Small Cell Lung Cancer	63 (28–76)				235 (60.4)		62 (31–83)				119 (60.7)	
O’Brien [9]	completely resected stage IB-IIIA non-small-cell lung cancer	65 (59.0–70.0)				285 (48%)	305 (52%)	65 (59.0–70.0)				273 (47%)	314 (53%)
		64.5 (60.0–69.5)				84 (50%)	84 (50%)	65.0 (58.0–71.0)				82 (50%)	82 (50%)
Kogure [10]	squamous non-small-cell lung cancer	76 (73–78)	61 (64%)	34 (36%)				77 (73–80)	65 (67%)	32 (33%)			
Peters [11]	metastatic NSCLC	66 (39–89)				108 (46)		66 (33–86)				102 (43)	
		65 (39–89)				72 (50)		66 (40–86)				65 (45)	
Westeel [12]	completely resected non-small-cell lung cancer	63.0 (56.7–70.5)											
Lu [13]	EGFR-mutated non-squamous non-small-cell lung cancer	59 (32–75)				104	44	57 (33–78)				115	36
Wang [14]	extensive-stage small-cell lung cancer	62 (55–66)				155 (67%)	75 (33%)	62 (56–67)				147 (63%)	85 (37%)
Saji [15]	small-sized peripheral non-small-cell lung cancer	67 (35–85)	211	341				67 (32–83)	211	343			
Forde [16]	Resectable Lung Cancer	64 (41–82)				93 (52.0)	86 (48.0)	65 (34–84)				83 (46.4)	96 (53.6)
Zhou [17]	metastatic non-small-cell lung cancer	62.0 (56.0–67.0)				202 (63%)	118 (37%)	64.0 (56.0–68.0)				91 (57%)	68 (43%)
Hellmann [18]	Advanced Non-Small-Cell Lung Cancer	64 (26–87)	58 (9.9)		219 (37.6)	306 (52.5)		64 (29–87)	55 (9.4)		223 (38.3)	305 (52.3)	

**Table 2 jpm-12-01998-t002:** Age grouping in clinical trials of drugs for breast cancer.

First Author/Reference	Disease	Age Groups in Treament Population					Age Groups in Ontrol/Other Treatment Population				
Age groups		Median	≥70	60–69	50–59	<50	Median	≥70	60–69	50–59	<50
Chua [19]	Non-low-risk ductal carcinoma in situ in the breast	58 (52–64)	74 (9%)	292 (36%)	306 (38%)	133 (17%)	57 (51–65)	71 (9%)	267 (33%)	334 (42%)	131 (16%)
Age groups		Median	≥40	<40			Median	≥40	<40		
Wang [20]	Metastatic triple-negative breast cancer	50 (22–69)	101 (79.5)	26 (20.5)			52 (30–75)	107 (85.0)	19 (15.0)		
Age groups		Median					Median				
Tripathy [21]	Metastatic Breast Cancer and Brain Metastases	53 (27–79)					52 (24–77)				
Age groups		Median	<65	≥65			Median	<65	≥65		
Xu [22]	Hormone receptor-positive and HER2-negative advanced breast cancer		211	30				108	12		
Age groups		Median	≥76	65–75	55–64	<55	Median	≥76	65–75	55–64	<55
Del Mastro [23]	Early-stage breast cancer	60 (54–67)	58	304	393	275	61 (54–68)	56	343	386	271
Age groups		median (IQR)	≥50	<50			median (IQR)	≥50	<50		
van der Voort [24]	ERBB2-Positive Breast Cancer	49 (43–55)	1101	118			48(43-56)	100	119		
Age groups		median (IQR)	>50	≤50			median (IQR)	>50	≤50		
Mayer [25]	Early breast cancer	52 (45–61)	1573	1309			52 (45–60)	1370	1304		
Age groups		Median age(IQR)	≤35	>35			Median age (IQR)	≤35	>35		
Yu [26]	Young Women With Breast Cancer	35 (32–38)	145 (55.6)	116 (44.4)			35 (31-37)	139 (53.5)	121 (46.5)		

**Table 3 jpm-12-01998-t003:** Age grouping in clinical trials of drugs for lymphoblastic leukemia.

First Author/Reference	Disease	Age Groups in Treatment Population						Age Groups in Control/Other Treatment Population					
		Median		≥10	1–10	<1		Median	≥10	1–10	<1		
Yang [27]	Acute lymphoblastic leukemia			21 (1.46)	1421 (98.54)	0			23 (1.55)	1458 (98.45)	0		
				263 (24.56)	777 (72.55)	31 (2.90)			265 (25.00)	758 (71.51)	37 (3.49)		
Age groups		Total		16–30	10–15	1–9		Analysis	≥16 Years	<16 Years			
Burke [28]	High-risk B-lymphoblastic leukemia	3040		20%	47%	33%			597	2443			
Age groups		Median			10–18	1–9		Median	10–18	1–9			
Locatelli [29]	High-risk First-Relapse B-Cell Acute Lymphoblastic Leukemia	6 (1–17)			15 (27.8)	39 (72.2)		5 (1-17)	16 (29.6)	38 (70.4)			
Age groups		Median (IQR)	21–27	18–20	13–17	10–12	1–9	Median (IQR)	21–27	18–20	13–17	10–12	1–9
Brown [30]	First Relapse of B-Cell Acute Lymphoblastic Leukemia	9 (6–16)	7 (6.7)	8 (7.6)	25 (23.8)	10 (9.5)	55 (52.4)	9 (5–16)	8 (7.8)	10 (9.7)	19 (18.4)	11 (10.7)	55 (53.4)
Age groups		Total		>14	>10–14	>6–10	4–6	Median		>14	>10–14	>6–10	4–6
Peters [31]	Childhood acute lymphoblastic leukemia	212		48 (23%)	64 (30%)	66 (31%)	34(16%)	201		62 (31%)	42 (21%)	75 (37%)	22 (11%)
Age groups				≥10	1–9				≥10	1–9			
Shen [32]	Pediatric Philadelphia Chromosome–Positive Acute Lymphoblastic Leukemia			60	123				X	X			
Age groups		Overall	≥10	<10					≥10	<10			
Place [33]	Newly diagnosed childhood acute lymphoblastic leukaemia	551	55 (24%)	176 (76%)					67 (29%)	165 (71%)			

## Data Availability

Not applicable.

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
