# Peer review of "A Historical Misconception in Clinical Trials of Drugs for Cancer—Age Grouping"

_jpm, 2022, doi:10.3390/jpm12121998_

Round 1

Reviewer 1 Report

The authors have discussed a pressing matter in current cancer clinical trials, emphasizing that other (sub)factors of patient immune, hormone systems and life stage could drive the drug response more than the age groups. The authors have used a good amount of related recent references and presented the views with clear data from referred trials.

However, the reviewer believes that more scientific data on life stage molecular (hormone, immune) response data should be presented and discussed elaborately.

Author Response

Dear Reviewer, we appreciate very much for your kindly reviewing of this manuscript.  We specially thank all your suggestions and comments, which are extremely helpful. Accordingly, we have addressed your comment below. We sincerely hope the manuscript reaches to the level for publication now.

  1. The authors have discussed a pressing matter in current cancer clinical trials, emphasizing that other (sub)factors of patient immune, hormone systems and life stage could drive the drug response more than the age groups. The authors have used a good amount of related recent references and presented the views with clear data from referred trials.

However, the reviewer believes that more scientific data on life stage molecular (hormone, immune) response data should be presented and discussed elaborately.

A:  Your point is important.   Although we have discussed these two aspects in some degree in different paragraphs, in considering your comments, we have added two paragraphs to address these two issues independently from other contents of the manuscript. In particular, these two subsections have been inserted into the section 2.

  2.7. Hormone levels and life stages

It is well known that the levels of hormones are associated with different life stages in both sexes [2].  In general, hormone levels increase during the life stage of body growth, maintain a relative high level during the reproductive stage, and de-crease at the aging stage. It is also known that the level of hormone is associated to the breast cancer and responding to the drug treatment of some cancers [22, 40-41]. As we discussed above, in many cases, the age grouping not agree with the life stages. There-fore, based on the relationship between levels of hormones and the disease development and response to drug treatment of the cancer patients, grouping with life sage will benefit the cancer treatment. 

2.8. Immune system and life stages

Similar to the hormone levels, the immune system develops with the body growth, keep functioning well during reproductive stage, and significantly decrease their levels when entering the aging stage. Immune levels are directly related to the cancer development and treatment [1-2]. The immune response and physiological status are different significantly among different life stages. Thus, immune system is highly relevant to the determination of treatment options and drug dosages of cancer patients. Life stage grouping can reflect correctly how the cancer patients response to the treatment.

In addition, we have gone through the manuscript to correct some minor English grammar errors.

Reviewer 2 Report

Dear Authors.

Please see minor comments for your submission below:

Abstract:

1.       Replace “In clinical trials of cancer clinical drugs” with In clinical trial of cancer drugs….

Introduction:

1.       Changing the second “the” to “an” in “Since the beginning of the cancer clinical trials, grouping patients is the essential part of the trial” would read better, i.e., is an essential part….

2.       “As personalized medicine depends heavily on the categorizing individuals into the right groups, aging groups in the cancer patients become one of the important issues to be solved.” This sentence is not clear. Please reword accordingly.

3.       Unfortunately, there is no systematically investigation on the relationship between the life stages

and the incidence of cancer types… Consider the use of “systematic” instead of “systematically”.

4.       Thus, it is most likely that incidence is low during the period between the puberty and menopause when the immune repose is high… It appears response is more appropriate here not repose.

5.       The immune system of the person after the menopause is weaker than that a person who did not reach to the menopause… Check sentence. It does not sound correct.

6.       The discovery of cancer drugs with the alkylating activity of nitrogen mustard [40-41]. All alkylating agents including nitrosourea Compounds, alkyl…. Please re-organize this section.

7.       Figure 2 is not labelled. It is referred to in the paragraph but not identified in the manuscript.

8.       5.1. Basic research to understand the similarities difference of cancer development and among life

Stages… Check this subtitle – “similarities difference”?

9.       Do section s 5.1, 5.2. and 5.3. form the conclusion to the manuscript?

Author Response

Dear Reviewer, we appreciate very much for your kindly reviewing of this manuscript.  We specially thank all your suggestions and comments, which are extremely helpful. Accordingly, we have addressed every comment below, and made additional changes whenever necessary in the manuscript. We sincerely hope the manuscript reaches to the level for publication now.

  1. Please see minor comments for your submission below:

Abstract:

  1. Replace “In clinical trials of cancer clinical drugs” with In clinical trial of cancer drugs….Done.

Introduction:

  1. Changing the second “the” to “an” in “Since the beginning of the cancer clinical trials, grouping patients is the essential part of the trial” would read better, i.e., is an essential part….Done.

  1. “As personalized medicine depends heavily on the categorizing individuals into the right groups, aging groups in the cancer patients become one of the important issues to be solved.” This sentence is not clear. Please reword accordingly.Done.

  1. Unfortunately, there is no systematically investigation on the relationship between the life stages

and the incidence of cancer types… Consider the use of “systematic” instead of “systematically”. Done.

  1. Thus, it is most likely that incidence is low during the period between the puberty and menopause when the immune repose is high… It appears response is more appropriate here not repose. Done.

  1. The immune system of the person after the menopause is weaker than that a person who did not reach to the menopause… Check sentence. It does not sound correct.Done.

  1. The discovery of cancer drugs with the alkylating activity of nitrogen mustard [40-41]. All alkylating agents including nitrosourea Compounds, alkyl…. Please re-organize this section.Done.

  1. Figure 2 is not labelled. It is referred to in the paragraph but not identified in the manuscript.Done.

  1. 5.1. Basic research to understand the similarities difference of cancer development and among life

Stages… Check this subtitle – “similarities difference”? Done.

  1. Do section s 5.1, 5.2. and 5.3. form the conclusion to the manuscript? Done.